# Recent Developments in Carbon-11 Chemistry and Applications for First-In-Human PET Studies

**DOI:** 10.3390/molecules28030931

**Published:** 2023-01-17

**Authors:** Anna Pees, Melissa Chassé, Anton Lindberg, Neil Vasdev

**Affiliations:** 1Azrieli Centre for Neuro-Radiochemistry, Brain Health Imaging Centre, Centre for Addiction and Mental Health (CAMH), Toronto, ON M5T 1R8, Canada; 2Institute of Medical Science, University of Toronto, Toronto, ON M5S 1A8, Canada; 3Department of Psychiatry, University of Toronto, Toronto, ON M5T 1R8, Canada

**Keywords:** Carbon-11, positron emission tomography (PET), radiochemistry, radiotracer, first-in-human

## Abstract

Positron emission tomography (PET) is a molecular imaging technique that makes use of radiolabelled molecules for in vivo evaluation. Carbon-11 is a frequently used radionuclide for the labelling of small molecule PET tracers and can be incorporated into organic molecules without changing their physicochemical properties. While the short half-life of carbon-11 (^11^C; t_½_ = 20.4 min) offers other advantages for imaging including multiple PET scans in the same subject on the same day, its use is limited to facilities that have an on-site cyclotron, and the radiochemical transformations are consequently more restrictive. Many researchers have embraced this challenge by discovering novel carbon-11 radiolabelling methodologies to broaden the synthetic versatility of this radionuclide. This review presents new carbon-11 building blocks and radiochemical transformations as well as PET tracers that have advanced to first-in-human studies over the past five years.

## 1. Introduction

Positron emission tomography (PET) is a molecular imaging technique that utilizes radiotracers for in vivo studies. The radionuclides fluorine-18 and carbon-11 are the most commonly used for labelling PET tracers because of the growing use of organofluorine drugs and as carbon is ubiquitous in nearly every drug or biomolecule. Additionally, their suitable decay characteristics and half-lives match the in vivo pharmacokinetics of small molecules. Consequently, developing new radiochemistry methods for the introduction of these short-lived radionuclides into organic molecules has emerged as one of the greatest challenges in PET radiopharmaceutical chemistry. Our ultimate goal is to radiolabel any molecule for medical imaging—a concept analogous to total synthesis that we introduced as “total radiosynthesis” [1]. Because the radiochemistry of fluorine-18 has been extensively reviewed in recent years [for example see: [2,3,4,5,6,7,8,9,10,11]], the focus of this review is on recent radiochemistry methodologies with carbon-11 and translation of ^11^C-labelled PET tracers to first-in-human (FIH) PET imaging studies. 

Carbon-11 (^11^C) has a half-life of 20.4 min and is produced in a cyclotron by proton bombardment of nitrogen gas in presence of trace amounts of oxygen (0.1–2%) or hydrogen (5–10%) where it is obtained as [^11^C]CO_2_ or [^11^C]CH_4_, respectively [12,13]. [^11^C]CO_2_ and [^11^C]CH_4_ can either be used directly in radiolabelling reactions or further converted to other ^11^C-building blocks (see Figure 1) [14]. The most common carbon-11 labelling strategy for PET tracers is ^11^C-methylation of hydroxy or amino groups using [^11^C]methyl iodide or [^11^C]methyl triflate, which are routinely obtained from [^11^C]CO_2_ and/or [^11^C]CH_4_. The advantages of ^11^C-methylation are the accessibility of precursors and carbon-11 methylating agents, as well as the general prevalence of methyl groups in pharmaceutical compounds. However, amongst molecules targeting the central nervous system (CNS) the prevalence of such methyl groups is rather low (<35%). Furthermore, metabolic demethylation can lead to cleavage of the radiolabel in vivo [15,16]. 

Synthetic efforts have been made in recent years to expand the toolbox for ^11^C-chemistry beyond ^11^C-methylation (Figure 1). Particular interest has been paid to the development of [^11^C]CO and [^11^C]CO_2_ chemistry, in order to gain access to ^11^C-labelled carbonyl-based functional groups. These radiochemistry methods open the door to labelling >75% of the compounds in CNS drug pipelines [15]. And other promising building blocks and synthetic strategies have been developed, such as new reactions with [^11^C]methyl iodide and related alklylating reagents, [^11^C]hydrogen cyanide, [^11^C]fluoroform, [^11^C]carbonyl difluoride, [^11^C]carbon disulfide, [^11^C]thiocyanate and [^11^C]formaldehyde (vide infra), further broadening the scope of compounds that can be labelled with carbon-11 and paving the way for our ultimate goal of total radiosynthesis. 

It should be noted that all yields are reported as they are stated or defined in their original articles (radiochemical yield (RCY), radiochemical conversion (RCC), radiochemical purity (RCP)) and might not necessarily reflect their definition as reported in the nomenclature guidelines [17,18]. The molar activity (A_m_) depends on several factors including the starting amount of radioactivity and is therefore difficult to compare.

## 2. Carbon-11 Methodologies

### 2.1. [^11^C]Carbon Dioxide

Historically, [^11^C]CO_2_ has been a challenging building block for radiochemists to use due to its moderate reactivity and potentially low A_m_ caused by isotopic dilution with atmospheric CO_2_. The introduction of bulky organic “fixation” bases such as 1,8-diazabicyclo [5.4.0]undec-7-ene (DBU) and 2-tert-butylimino-2-diethylamino-1,3-dimethylperhydro-1,3,2-diazaphosphorine (BEMP) for trapping of [^11^C]CO_2_ [19,20,21] was inspired by green chemistry for capturing atmospheric CO_2_ and represents a major advance for ^11^C-chemistry: the “fixation” bases allow [^11^C]CO_2_ to be easily trapped in a reaction vessel at room temperature and enable access to high oxidation state functional groups such as carbon-11 labelled carboxylic acids, amides, formamides, ureas, carbamates and other functional groups [22]. This methodology has contributed to the accessibility of [^11^C]CO_2_ as a building block and, in consequence, a wide array of new [^11^C]CO_2_ chemistry applications and PET tracers have emerged over the past decade. This review will focus on novel [^11^C]CO_2_ fixation reactions reported within the last five years. 

While [^11^C]CO_2_ is directly produced in the cyclotron, the irradiated cyclotron target gas contains many undesired chemical and radiochemical entities. To purify [^11^C]CO_2_ from carrier gases and other by-products, it is typically trapped using liquid nitrogen or by physical adsorption on porous polymers, such as carbon molecular sieves or polydivinylbenzene copolymers. A new method for purifying [^11^C]CO_2_, also inspired by green chemistry literature, has recently been reported by our laboratories which employs chemisorption by solid polyamine-based adsorbents. This method uses small amounts of silica-grafted polyethyleneimine to trap [^11^C]CO_2_ at room temperature and quantitatively release it under mild heating (85 °C). Trapping efficiencies (TEs) as high as 79 ± 12% were observed but decreased over multiple cycles, indicating a limited reusability of the capture material. This technology was applied to synthesize a PET tracer by [^11^C]CO_2_ fixation reactions, and could potentially be applied for solid phase reactions as well as enable the transportation of carbon isotopes [23]. 

Traditionally, direct use of [^11^C]CO_2_ is achieved by use of Grignard reagents, organolithiums or silanamines to yield ^11^C-labelled amides or carboxylic acids. However, these reagents are challenging to implement in automated PET tracer production due to their hygroscopic nature, tendency to absorb atmospheric CO_2_, and corrosiveness [22]. As such, many new methodologies for the preparation of [^11^C]carboxylic acids have been developed over the past five years by novel [^11^C]CO_2_ fixation reactions that employ the aforementioned “fixation” bases. Our laboratory reported the use of aryl and heteroaryl stannanes as precursors which were carboxylated in a copper(I)-mediated reaction with [^11^C]CO_2_ (see Figure 2A) [24]. The method was fully automated and applied for an alternative synthesis of [^11^C]bexarotene (previously synthesized by reaction of [^11^C]CO_2_ with a boronic ester precursor mediated by a copper(I) source [25,26]), and was obtained with a RCY of 32 ± 5% (decay-corrected (dc)) and a A_m_ of 38 ± 23 GBq/µmol. The strategy was also applied by García-Vázquez et al. to the synthesis of ^11^C-carboxylated tetrazines for the labelling of trans-cyclooctene-functionalized PeptoBrushes [27]. After optimization of the original reaction conditions (CuI instead of CuTC, NMP instead of DMF and addition of TBAT as fluoride ion source), two tetrazines were successfully ^11^C-carboxylated with RCYs of 10–15% and “clicked” to the TCO-PeptoBrushes. It is noteworthy that Goudou et al. reported the copper-catalyzed radiosynthesis of [^11^C]carboxylic acids by reaction of [^11^C]CO_2_ with terminal alkynes in presence of DBU (see Figure 2B) [28]. A small library of [^11^C]propiolic acids was obtained with RCYs between 7 and 28%. A different approach using trimethyl and trialkoxy silanes as precursor has been described by Bongarzone et al. (see Figure 2C) [29]. In this desilylative carboxylation reaction, aromatic silane precursors were activated by fluoride, forming a pentavalent silicate which was then reacted in a copper-catalyzed reaction with [^11^C]CO_2_. [^11^C]Carboxylic acids were obtained with RCYs of 19–93% and TEs of 21–89%. A more general approach for the synthesis of [^11^C]carboxylic acids was introduced by Qu et al. (see Figure 2D) [30]. Sp-, sp^2^- and sp^3^-hybridized carbon-attached trimethylsilanes were ^11^C-carboxylated in a fluoride-mediated desilylation (FMDS) reaction, resulting in a broad substrate scope and high RCYs (up to 98%). The applicability of the method was demonstrated by synthesizing two carboxylic acid PET tracers via the FMDS approach. 

[^11^C]Carboxylic acids can also be synthesized by isotopic exchange reactions. Destro et al. reported the isotopic exchange reaction of cesium salt precursors with ^13^C, ^14^C, and a few selected examples of ^11^C (see Figure 3A) [31]. While good yields were obtained for [^13^C]CO_2_ and [^14^C]CO_2_, yields were low for [^11^C]CO_2_ (3–50%) due to low TEs. Another take on this strategy was demonstrated by Kong et al., who employed photoredox catalysis and obtained similar results (see Figure 3B) [32]. In both cases, A_m_ was low, as expected (<0.2 GBq/µmol). A very recent addition to the portfolio of carboxylic acid labelling strategies by isotopic exchange was presented by Bsharat et al. [33]. These authors developed an aldehyde-catalyzed carboxylate exchange reaction in α-amino acids (see Figure 3C) with ^13^C and ^11^C. For the ^11^C-reactions, imine carboxylates were pre-formed by condensation of α-amino acids with aryl aldehydes and subsequently subjected to the carboxylate exchange reaction with [^11^C]CO_2_. An array of α-amino acids was labelled with RCYs of 4–24%, and the modest yields were also attributed to low TEs of the [^11^C]CO_2_. Phenylalanine was isolated by this reaction with a A_m_ of 8.4 GBq/mmol. 

Figure 4 gives an overview of the proposed mechanism of [^11^C]CO_2_ fixation with fixation bases such as BEMP and DBU and formation of the [^11^C]isocyanate, as well as the ^11^C-labelled functional groups that can be obtained via this pathway [18]. While early works focused on the synthesis of carbamates, the scope of ^11^C-labelled functional groups has broadened immensely over time. 

The efficient syntheses of carbon-11 labelled amides, ureas, and formamides have been a longstanding goal in PET radiochemistry and have seen an emergence of interest in recent years. Bongarzone et al. reported a rapid one-pot synthesis of amides via a Mitsunobu reaction (see Figure 5A) [34]. [^11^C]CO_2_ was trapped with DBU, converted to [^11^C]isocyanate (or an [^11^C]oxyphosphonium intermediate) using Mitsunobu reagents and subsequently reacted with a Grignard reagent to form the respective amide. RCYs of up to 50% were obtained. The substrate scope was not investigated for [^11^C]CO_2_, but [^11^C]melatonin was synthesized to demonstrate the applicability of this method to biologically relevant compounds. Mair et al. used organozinc iodides as alternatives to Grignard reagents in a rhodium-catalyzed addition to [^11^C]isocyanates (see Figure 5B) [35]. The isocyanates were generated similarly to the previous method and reacted with the organozinc iodides in presence of a rhodium catalyst with RCYs of 5–99%. One model compound was isolated with a RCY of 12% and A_m_ of 267 GBq/µmol to demonstrate suitability for PET tracer production. In order to develop an efficient synthesis strategy for the benzimidazolone PET tracer (*S*)-[^11^C]CGP12177, Horkka et al. reported a BEMP/Mitsunobu-based strategy for the synthesis of cyclic aromatic ureas: *ortho*-Phenylenediamines were reacted with [^11^C]CO_2_ in presence of BEMP as fixation base. Mitsunobu reagents (DBAD, nBu_3_P) were added to form the [^11^C]isocyanate intermediates which then reacted intramolecularly to yield the respective ^11^C-labelled urea (see Figure 5C) [36]. The strategy was also applied to cyclic carbamates and thiocarbamates, as well as the tracer (*S*)-[^11^C]CGP12177, which was obtained in 23% RCY (dc) with a A_m_ of 14 GBq/µmol. Luzi et al. reported the synthesis of [^11^C]formamides (see Figure 5D) [37]. [^11^C]CO_2_ was trapped with BEMP in diglyme and was reacted with aromatic and aliphatic primary amines to form the respective [^11^C]isocyanates, which were subsequently reduced to the [^11^C]formamides with sodium borohydride. The method performed better for aliphatic amines compared to aromatic amines.

In an attempt to make [^11^C]CO_2_ fixation with BEMP and DBU more widely accessible and amenable to automation, two strategies of “in-loop” [^11^C]CO_2_ fixation have been developed. While our laboratory developed this method using a standard stainless-steel HPLC loop for [^11^C]CO_2_ fixation, Downey et al. applied a disposable ethylene tetrafluoroethylene loop [38,39]. In both cases, [^11^C]CO_2_ was captured in the loop in the presence of an amine precursor and fixation base, prior to reaction with a model substrate. The “in-loop” fixation has been applied to synthesize ^11^C-labelled carbamates, unsymmetrical, and symmetrical ureas. 

A different approach to access ureas and carbamates via [^11^C]isocyanate was presented by Audisio and co-workers (see Figure 6). The [^11^C]isocyanate intermediates were generated through a Staudinger aza-Wittig reaction from the respective azide, then reacted either intramolecularly to form cyclic [^11^C]ureas [40] and [^11^C]carbamates [41] or intermolecularly with an amine to form linear ureas [42]. All three strategies were applied for the synthesis of ^13^C-, ^14^C- and ^11^C-labelled compounds. RCYs of the isolated ^11^C-compounds generally ranged between 20 to 50%.

To avoid the multi-step syntheses and limited substrate scope of previously reported methods, Liger et al. reported a novel radiolabelling strategy for benzimidazoles and benzothiazoles (see Figure 7). In this work, [^11^C]CO_2_ was reacted with aromatic diamines and aminobenzenethiols in presence of 1,3-bis(2,6-diisopropylphenyl)imidazol-2-ylidene (IPr), zinc chloride, and phenylsilane as reducing reagent to obtain various benzimidazoles and benzothiazoles [43]. 

Previously, the synthesis of [^11^C]carbonates could only be achieved using the esoteric building block [^11^C]phosgene, which is technically challenging to prepare and requires specialized apparatus. To access this functional group directly from [^11^C]CO_2_, Dheere et al. developed a procedure involving an alkyl chloride, an alcohol, TBAI and base in DMF (see Figure 8) [44]. The procedure was used for the synthesis of one model compound, and resulted in either moderate RCY (31 ± 2%) and higher A_m_ (10–20 GBq/µmol; low amounts of ^11^C), or high RCY (up to 82%) and lower A_m_ (2 GBq/µmol), depending on the base. 

A novel method for ring-opening non-activated aziridines with [^11^C]CO_2_ using DBU/DBN halide ionic liquids was developed by our laboratory (see Figure 9) [45]. [^11^C]CO_2_ was introduced to a pre-activated mixture of benzyl aziridine and the ionic liquid giving 4-benzyl [^11^C]oxazolidine-2-one with 77% radiochemical conversion (RCC) and 78% TE. The method was applied to radiolabel an array of [^11^C]oxazolidinones (RCCs 5–95%) as well as a MAO-B inhibitor, [^11^C]toloxatone, as a proof of concept. 

### 2.2. [^11^C]Carbon Monoxide

[^11^C]Carbon monoxide has gained much interest in recent years. Novel ^11^CO-chemistry will not be covered within this review but we refer to recent comprehensive reviews of [^11^C]CO production methods and ^11^C-carbonylation chemistry [46,47,48,49,50]. Although many straightforward routes for [^11^C]CO production have been established, and a diverse portfolio of [^11^C]carbonylation reactions has been developed, this branch of carbon-11 chemistry is still heavily underrepresented in PET tracer synthesis. In fact, of the 100+ labelled compounds synthesized from [^11^C]CO, only four are reported for human use to our knowledge [51]. One likely reason for the hampered translation of [^11^C]CO radiochemistry to the clinic can be attributed to the historic lack of commercially available automated synthesis units for [^11^C]CO. This has now been overcome with systems such as the TracerMaker^TM^ which is used by our laboratories for the syntheses of *N*-[^11^C]acrylamide PET tracers for imaging Bruton’s tyrosine kinase via a palladium-NiXantphos-mediated carbonylation using [^11^C]CO [51,52]. The synthesis of the same class of compounds has also recently been automated as “in-loop” procedure using the GE TracerLab synthesis modules [53]. Prior to this recent work, ^11^C-labelled *N*-acrylamides were synthesized from [^11^C]acrylic acid or [^11^C]acryloyl chloride (formed by carboxylation of Grignard or organolithium reagents with [^11^C]CO_2_) and were not suitable for human translation.

### 2.3. [^11^C]Methyl Iodide and Other ^11^C-Alkylation Agents

[^11^C]Methyl iodide and [^11^C]methyl triflate have been known for many decades [54,55,56,57] and are by far the most commonly used ^11^C-labelling agents. Their widespread use is attributed to their routine radiosyntheses and high reactivity. Both [^11^C]methyl iodide and [^11^C]methyl triflate can be easily synthesized from the primary cyclotron products (i.e., [^11^C]CH_4_ or [^11^C]CO_2_) using the classical wet-chemistry approach with lithium aluminium hydride and HI or the gas-phase method involving I_2_, and dedicated synthesis devices with fully automated procedures are commercially available [58]. Mostly, [^11^C]methyl iodide and [^11^C]methyl triflate are employed in ^11^C-methylation reactions of hydroxyl, amine or thiol precursors, but also many different ^11^C-C coupling reactions have been established, including Suzuki, Stille, and Negishi couplings. For an overview of ^11^C-C cross-coupling strategies, we refer the reader to a comprehensive review from H. Doi [59]. Recent progress in the field has been made by Rokka et al., who systematically studied the reaction of various organoborane precursors with [^11^C]methyl iodide in two different reaction media, DMF(/water) and THF/water, to determine the best precursor and solvent for Suzuki-type cross coupling reactions in ^11^C-chemistry [60]. These authors found that for their model compound (1-[^11^C]methylnaphthalene), the boronic acid and pinacol ester precursors gave the highest yields, while the solvent mixture THF/water was equal or superior in any tested reaction. Recent work focused on broadening the substrate scope to diversify ^11^C-methylation chemistry. Pipal et al. reported the ^11^C-methylation of aromatic and aliphatic bromides via metallaphotoredox catalysis (see Figure 10A) [61]. The applicability of this labelling strategy was demonstrated by synthesizing 11 ^11^C-labelled biologically active compounds, including the PET tracers [^11^C]UCB-J and [^11^C]PHNO, in RCYs of 13–72% for proof of concept. Qu et al. extended their fluoride-mediated desilylation of organosilanes, initially developed for [^11^C]CO_2_ fixation (vide supra), to [^11^C]methyl iodide and succeeded in labelling a diverse library of silane substrates with RCYs of up to 93% (see Figure 10B) [25]. 

As an alternative to [^11^C]CO or [^11^C]acetyl chloride chemistry, Dahl and Nordeman developed a procedure for ^11^C-acetylation of amines with [^11^C]methyl iodide (see Figure 10C) [62]. Bis(cyclopentadienyldicarbonyliron) was used as the CO source in the Pd-mediated reaction. The reaction was established for a range of primary amine precursors, including three biologically relevant compounds, and a few examples of secondary amines. A different approach to the same functional group was presented by Doi et al., whereby [^11^C]acetic acid was synthesized in a palladium-mediated cross-coupling reaction from [^11^C]methyl iodide and carboxytriphenylsilane, then converted to the [^11^C]acetic acid phthalimidyl ester or succinimidyl ester (see Figure 10D) [63]. The imidyl esters were subsequently employed in a ^11^C-acetylation reaction with small, medium-sized, and large molecules. 

In an effort to develop a stereoselective ^11^C-alkylation procedure for diastereomerically enriched dipeptides, Filp et al. investigated the use of various quaternary ammonium salts as chiral phase-transfer catalysts in the ^11^C-alkylation of *N*-terminal glycine Schiff bases (see Figure 10E) [64]. Next to [^11^C]methyl iodide, the procedure was also applied to [^11^C]benzyl iodide. RCCs of >80% and high diastereomeric ratios (d.r.) of up to 95:5 were obtained. A similar strategy has been used by Pekošak et al. for the stereoselective ^11^C-labelling of the tetrapeptide Phe-D-Trp-Lys-Thr with [^11^C]benzyl iodide [65]. [^11^C]Phe-D-Trp-Lys-Thr was synthesized over five steps starting from [^11^C]CO_2_ and isolated with high stereoselectivity (94% de), RCYs of 9–10% (dc), and A_m_ of 15–35 GBq/µmol. 

To address the shortcomings of current cross-coupling strategies with [^11^C]methyl iodide, Helbert et al. developed a new cross-coupling procedure with [^11^C]methyllithium. [^11^C]Methyllithium was developed as a more reactive alternative for [^11^C]methyl iodide and can be synthesized by reaction of [^11^C]methyl iodide with *n*-butyllithium [66,67]. In the procedure of Herbert et al., [^11^C]methyllithium was added without intermediate purification to the aryl bromide precursors and a selection of relevant PET tracers was labelled by palladium-mediated ^11^C-C cross-coupling with RCYs of 33–57% (see Figure 11) [68].

### 2.4. [^11^C]Hydrogen Cyanide

Since its inception in the 1960s [69], [^11^C]HCN has developed into a versatile building block for the ^11^C-labelling of neurotransmitters, amino acids, and other molecules. This is mainly due to its versatility: It can function as nucleophile as well as electrophile, and [^11^C]cyanide incorporation generates many different functionalities, such as nitriles, hydantoins, (thio)cyanates and, through subsequent reaction, carboxylic acids, aldehydes, amides and amines. Two extensive reviews on [^11^C]hydrogen cyanide have been recently published, therefore this ^11^C-building block will not be discussed in detail herein [70,71]. Since [^11^C]hydrogen cyanide is one of the few ^11^C-building blocks used for FIH PET tracers in recent years (vide infra), we will provide a brief summary of recent work that has not been covered by other reviews.

[^11^C]Hydrogen cyanide is typically produced by reacting [^11^C]CH_4_ with NH_3_ gas on a platinum catalyst at 1000 °C. While fully automated production systems are commercially available, [^11^C]HCN is not widely used. In an effort to make [^11^C]hydrogen cyanide more accessible, Kikuchi et al. developed a novel synthesis strategy from widely available [^11^C]methyl iodide (see Figure 12) [72]. This method involves passing [^11^C]methyl iodide over a heated reaction column, in which it is first converted to [^11^C]formaldehyde and subsequently to [^11^C]hydrogen cyanide. The [^11^C]hydrogen cyanide is obtained fast and with RCYs comparable to the traditional method (50–60% at EOB), without the need for specialized equipment. 

### 2.5. [^11^C]Fluoroform

Due to the prevalence of CF_3_ groups in drugs and other biologically active compounds, there has been much interest in labelling this group with carbon-11 and fluorine-18. Haskali et al. published a synthesis procedure for carbon-11 labelled fluoroform in 2017, where cyclotron-produced [^11^C]methane was fluorinated by passing it over a CoF_3_ column at elevated temperatures (270 °C) [73]. [^11^C]Fluoroform was obtained with RCYs of ~60%. The process was not only fast and reproducible, but the developed system also required very little maintenance. [^11^C]Fluoroform was reacted with various model compounds (see Figure 13), in addition to three biologically active compounds.

Whereas flourine-18 labelled fluoroform generally suffers from low molar activities (≤1 GBq/µmol) and only few examples of higher molar activities are known, high molar activities of >200 GBq/µmol were easily obtained with carbon-11 labelled fluoroform. In later works, the substrate scope of reactions with [^11^C]fluoroform was broadened from aryl boronates, aryl iodides, ketones, diazonium salts, and diarylsulfanes to aryl amines and arylvinyl iodonium tosylates (see Figure 13) [74,75]. 

### 2.6. [^11^C]Carbonyl Difluoride

As an alternative strategy to access carbon-11 labelled ureas, carbamates, and thiocarbamates, Jakobsson et al. presented the [^11^C]carbonyl group transfer agent [^11^C]carbonyl difluoride [76]. [^11^C]Carbonyl difluoride was synthesized quantitatively by passing [^11^C]CO over a AgF_2_ column at room temperature. The building block was subsequently reacted with diamines, aminoalcohols, and aminothiols to form the corresponding cyclic azolidin-2-ones (see Figure 14) under mild conditions with very low precursor quantities, and even in presence of water. The same laboratory expanded their procedure to linear unsymmetrical ureas and established reaction conditions for a broad scope of aryl and aliphatic amines [77]. For the aryl amines, [^11^C]carbonyl fluoride was trapped in a solution with the aryl amine precursor and subsequently reacted with another amine. For the aliphatic amines, alkylammonium tosylate precursors were used in the first step to lower the reactivity of the amine and prevent symmetrical urea formation. Pyridine was used to improve [^11^C]carbonyl fluoride trapping. Suitability for PET tracer synthesis was demonstrated by labelling the epoxide hydrolase inhibitor [^11^C]AR-9281, which was obtained after optimization in high RCYs of 80%.

### 2.7. [^11^C]Carbon Disulfide

[^11^C]Carbon disulfide, the sulfur analog of [^11^C]carbon dioxide, is an interesting ^11^C-building block for the synthesis of organosulfur compounds. It has first been described in 1984, and had limited utility until a decade ago when Miller and Bender proposed a new synthesis strategy, which was further improved by Haywood et al. [78,79,80]. It can now be readily obtained through the reaction of [^11^C]CH_3_I with elemental sulfur and has been used to synthesize [^11^C]thioureas, thiocarbamates and related structures. Cesarec et al. recently published a procedure for the synthesis of late transition metal complexes with [^11^C]dithiocarbamate ligands [81]. To this end, [^11^C]carbon disulfide was reacted with diethyl amine or dibenzyl amine to form the respective ammonium [^11^C]dithiocarbamate salt and subsequently reacted with Au(I), Au(III), Pd(II) or Pt(II) complexes to form the respective complexes in RCYs > 70% (see Figure 15). 

### 2.8. [^11^C]Thiocyanate

[^11^C]Thiocyanate is an interesting ^11^C-building block because of its reactivity and potential to give access to a wide range of organosulfur derivatives. Up until recently, its production relied on the use of [^11^C]HCN [82]. Haywood et al. presented a new way to synthesize this ^11^C-building block by reacting [^11^C]carbon disulfide with ammonia at 90 °C to form ammonium [^11^C]thiocyanate in near quantitative RCC [83]. The ammonium [^11^C]thiocyanate was subsequently reacted with benzyl bromide, a range of α-ketobromides, and mannose triflate in high RCYs of ≥75% (see Figure 16). The α-[^11^C]thiocyanatophenones could also be cyclized in the presence of sulfuric and acetic acid to ^11^C-thiazolones. 

### 2.9. [^11^C]Formaldehyde

[^11^C]Formaldehyde is an established and versatile building block for carbon-11 chemistry (see [84] and references therein). Many different synthetic strategies have been proposed, traditionally involving reduction in cyclotron-produced [^11^C]CO_2_ to [^11^C]CH_3_OH and subsequent oxidation to [^11^C]formaldehyde. Nader et al. recently proposed the use of XeF_2_ as an oxidizing agent (see Figure 17) [85]. [^11^C]Formaldehyde was obtained in non-decay corrected RCYs of 54 ± 5% starting from [^11^C]CO_2_ and was used in a proof-of principle synthesis to form α-(*N*-[^11^C]methylamino)isobutyric acid via reductive ^11^C-methylation. 

## 3. First-in-Human Translation

Despite the short physical half-life and the need for an on-site cyclotron, ^11^C continues to be a favoured radionuclide for small molecule PET tracers. As discussed in this review, innovations continue in ^11^C-radiolabelling strategies for applications in ^11^C-tracer development. Within the past five years, to our knowledge at least 27 novel ^11^C-labelled PET tracers have been translated for FIH PET studies (see Figure 1) [86,87,88,89,90,91,92,93,94,95,96,97,98,99,100,101,102,103,104,105,106,107,108,109]. Unsurprisingly, the vast majority of these PET tracers were designed to image targets within the CNS (see Table 1). Carbon-11 is ideal for CNS PET because the substitution of naturally occurring ^12^C with ^11^C does not change the physicochemical properties of the compound, thereby enabling imaging with isotopologues of the molecules of interest for accurate determination of brain penetrance, target affinity, pharmacokinetics, or pharmacodynamics of the molecule, and multiple scans can be performed in the same subject in the same day. Interestingly, many of the ^11^C-labelled PET tracers for FIH use focused on imaging markers of neuroinflammation, a critical component in the etiology and pathology of several neurodegenerative diseases, including Alzheimer’s disease (AD), Parkinson’s disease (PD), and amyotrophic lateral sclerosis (ALS) [110,111,112,113,114]. The remaining PET tracers translated for FIH studies that were reported in the past five years strived to image non-CNS targets, including bacterial infection and lung inflammation. 

When sorting the tracers according to the labelling method, it becomes immediately apparent that the predominant synthetic strategy remains ^11^C-methylation of hydroxy or amino precursors: more than ¾ of all tracers were synthesized via this strategy, either using [^11^C]methyl iodide or [^11^C]methyl triflate as the ^11^C-buidling block (see Figure 2). This can be attributed to the accessibility of these building blocks from cyclotron-produced [^11^C]CO_2_ or [^11^C]CH_4_ and the availability of commercial synthesis devices (vide supra) [58]. Other tracers have been synthesized by alternate ^11^C-labelling strategies, using [^11^C]HCN or Grignard reactions with [^11^C]CO_2_. The latest developments in ^11^C-chemistry are not represented among the FIH tracers, which is not surprising since it usually takes time for a new method to be implemented by the broader community. However, many of the existing ^11^C-building blocks have not been introduced in the past few years but have been around for decades and should, therefore, be available for clinical application. As indicated in some cases (e.g., [^11^C]CO chemistry), it may be the historic lack of specialized or commercially available radiosynthesis equipment that hampers FIH translation of new PET tracers. Other reasons could be that new ^11^C-methodologies are often only developed up to the point of proof-of-principle and not optimized for automated tracer production. Rather than further broadening the scope of ^11^C-chemistry, future efforts should focus on closing the gap between new method development and clinical translation.

## Data Availability

Not applicable.

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
