# Peer review of "Recent Developments in Carbon-11 Chemistry and Applications for First-In-Human PET Studies"

_molecules, 2023, doi:10.3390/molecules28030931_

Round 1

Reviewer 1 Report

The manuscript - Recent developments in carbon-11 chemistry and applications for first-in-human PET studies - by Pees et al. fill a gap spreading the knowledge of important radiochemical achievements to a broader audience. The importance of carbon-11 as a research tool cannot be overestimated, which is nicely described and argued why in the manuscript. The examples of new radiochemistry approaches are well selected, and the manuscript covers most of the carbon-11 chemistry through the references and pointing out important complementary review articles. 

There are a few minor comments which the authors could consider.

1. The reference policy is not fully to my taste. I suggest that in a review article, important or interesting achievements from the past should be acknowledged, even if the focus is "recent developments". Those who read this manuscript might get the impression that some work from recent years was the original discovery. I leave to the authors and the editor to decide on this, but give a few examples;

The use of 11C-methyl lithium was described in NMB (1994) 21:1067-1072 (perhaps also earlier); Carbon-11 disulfide at least 1984 IJARI 35:29-33, and carbon-11 methyl iodide independently by Langstrom et al and Comar et al 1976. There are other examples.

2. Comments on the text

General comments: To be consistent, remove all notation about decay corrected or not and same applies for RCY “HPLC”. The authors have stated in the beginning that all RCY are taken directly as written in the original publication. Maybe the authors should mention that the molar activity depends on starting amount of radioactivity, and therefore difficult to compare between published data without detailed knowledge of the experimental set up. The use of high or low when commenting molar activity should thus be avoided except for very high (maybe >200 GBq/umol) and very low (<5 GBq/uml) or if there is a scientific reason of interest behind.

In the abstract the uthors use ….radiolabelled biomolecules… This is rather limiting since biomolecules by definition means “compounds produced by living organism”, molecules would be more accurate. They also use radiopharmaceuticals as a general term for radiolabelled compounds. Radiopharmaceuticals are compounds used for clinical diagnosis and it would be better to use one notation for all labelled compounds, such as radiotracer, PET-tracers etc.

Line 5 in Introduction: …good decay characteristics and suitable half-lives. Suggest ….suitable decay characteristics and half-lives…..

Page 2 first line: ……..biologically active compounds. Maybe “active” could be changed to interesting or useful, since active may infer some biological effect.

On page 7 below Fig. 8 [11C]4-benzyl oxazolidine-2-one should be written 4-benzyl [11C]oxazolidine-2-one

Page 13 2. First in-human-translation line 12: 11C-synthons should be replaced by 11C-precursors (synthons is not equivalent to a certain compound, “a structural unit within a molecule that can be formed by known synthetic operations”).

The text in the end of the same paragraph is repeated on page 16 “Other reasons could be that new 11C-methodologies…… 

Reviewer 2 Report

This is a very well written review on new advances in carbon-11 radiochemistry.

The authors have included new methods from the literature covering from 2017 to today.

The methods are understandably very briefly described. However, because many of these are complicated in the reagents and the reaction mechanisms cannot be well understood by readers who are not experts in carbon-11 radiochemistry, like for example the reaction described in Scheme 10A.

Also, adding an abbreviation section would be useful, as the majority of terms are abbreviated. 
